# Influence of Non-*Saccharomyces* Strains on Chemical Characteristics and Sensory Quality of Fruit Spirit

**DOI:** 10.3390/foods10061336

**Published:** 2021-06-10

**Authors:** Fatjona Fejzullahu, Zsuzsanna Kiss, Gabriella Kun-Farkas, Szilárd Kun

**Affiliations:** 1Department of Bioengineering and Alcoholic Drink Technology, Institute of Food Science and Technology, Hungarian University of Agriculture and Life Sciences, H-1118 Budapest, Hungary; fatjona.fejzullahu@uni-pr.edu (F.F.); kiss.zsuzsanna@uni-mate.hu (Z.K.); kun-farkas.gabriella@uni-mate.hu (G.K.-F.); 2Department of Food Technology, Faculty of Agriculture and Veterinary, University of Prishtina, 10000 Prishtina, Kosovo

**Keywords:** non-*Saccharomyces* yeasts, mixed fermentation, fruit spirit, volatile compounds, sensory attributes, quality

## Abstract

The use of non-*Saccharomyces* yeasts for alcoholic beverage improvement and diversification has gained considerable attention in recent years. The effect of pure and mixed inocula (of *Torulaspora delbrueckii*, *Lachancea thermotolerans*, and *Saccharomyces cerevisiae*) on apple mash fermentation has been determined for the production of Hungarian fruit spirit (Pálinka), with a special emphasis on the chemical, volatile, and sensory attributes. The enological parameters were followed during the fermentation process. Sugar consumption and organic acid production were determined by HPLC, whereas the aromatic profile of the distillates was characterized by GC-FID. According to the results, single and mixed cultures showed similar characteristics during mash fermentation. The identified volatile compounds included aldehydes, esters, and higher alcohols. Mixed culture fermentation trials revealed a significantly higher concentration of volatile compounds and better sensorial attributes compared to those exhibited by the pure culture of *S. cerevisiae*.

## 1. Introduction

Alcoholic fermentation is a complex biochemical process performed by yeasts that utilize sugars and other constituents as substrates for their metabolism, converting these to ethanol, carbon dioxide, and other metabolic byproducts that contribute to the chemical composition and quality of the beverage [1]. Studies have shown that non-*Saccharomyces* are the most prevalent yeast genera in the first stages of spontaneous and inoculated fermentation, while *S. cerevisiae* strains are dominant during the latter stages [2,3]. The causes underlying yeast interactions during fermentation are not fully understood. According to Nissen et al. [4], the early growth arrest of non-*Saccharomyces* yeasts (*L. thermotolerans* and *T. delbrueckii*) is not due to the presence of ethanol or other toxic compounds, but it seems to be triggered by a cell–cell contact mechanism dependent on the presence of viable *S. cerevisiae* cells at high concentrations. Earlier studies considered non-*Saccharomyces* yeasts as ‘wild’ or ‘spoilage’ yeasts because they were often isolated from stuck or sluggish fermentations, or wines with anomalous analytical and sensorial profiles. However, the total suppression of indigenous non-*Saccharomyces* species can diminish the aroma complexity of the final beverage.

In this context, it has been proposed that non-*Saccharomyces* yeast strains should be included in mixed and multi-starter cultures alongside *Saccharomyces* strains to improve the chemical composition and sensory properties of alcoholic beverages, while avoiding the undesirable compounds that these species might produce [5]. The positive impact of multi-starter fermentation on the complex flavor and quality of the wine [6,7], tequila [8], and sugar cane spirit has been reported recently [9,10]. Therefore, the aim of the present study was to investigate the fermentation activity of non-*Saccharomyces* strains in apple mash and to determine their effect on the quality of Pálinka.

## 2. Materials and Methods

### 2.1. Raw Materials, Yeast Strains, and Chemicals

Apples (*Malus domestica* ‘Jonathan’, Csány 1) were collected beside the city of Székesfehérvár, located in central Hungary, in September 2019, and were transported to the laboratory for subsequent analysis. The Jonathan apple is a classic American variety, medium-sized apple, and widely regarded as one of the best flavored, with a good sweet/sharp balance. The Viniflora Concerto^TM^ (*L. thermotolerans*) and Melody^TM^ (mixed culture of *T. delbrueckii*, *L. thermotolerans*, and *S. cerevisiae*) starter cultures were obtained from Chr. Hansen A/S (Hoersholm, Denmark), while the Biodiva^TM^ Level 2 (*Torulaspora delbrueckii* TD291) and Uvaferm 228 (*S. cerevisiae*) of Lallemand Inc. (Montréal, QC, Canada) dry yeasts were purchased from Kokoferm Ltd. (Gyöngyös, Hungary). Standards (glucose, fructose, saccharose, acetic acid, lactic acid, succinic acid, acetaldehyde, methanol, isoamyl alcohol, 1-propanol, 1-butanol, 2-butanol, 2-phenylethanol, 1-hexanol, 2-methyl-1-butanol, trans-3-hexen-1-ol, cis-2-hexen-1-ol, benzyl alcohol, ethyl acetate, propyl acetate, ethyl hexanoate, ethyl butyrate, isoamyl acetate, phenylethyl acetate, diethyl succinate, ethyl octanoate, ethyl benzoate, ethyl formate, linalool), and all chemicals of analytical grade were obtained from Sigma-Aldrich (Steinheim, Germany).

### 2.2. Mashing and Fermentation Conditions

The purchased apples were sorted (mechanically harmed, decayed, and rotten fruits were excluded) and cleaned gently with water to remove dust and debris. The fruits were roughly crushed using a grinder and then placed in 50 L stainless-steel fermentation tanks, each containing 35 kg of mash. The pH of the mash was adjusted to 3.0 using phosphoric and lactic acid in a ratio of 95:5. Lallzyme^TM^ HC (Lallemand, Montréal, QC, Canada) enzyme preparation was used at a dose of 3 g/100 kg to decompose the pectin. Thereafter, 20 g/100 kg Uvavital^TM^ (Lallemand, Montréal, QC, Canada) yeast nutrient was added to each tank. Fermentation was initiated by adding rehydrated yeast starter. *T. delbrueckii* and *L. thermotolerans* were used in sequential inoculation with *S. cerevisiae*. The mash was first inoculated with the non-conventional yeast at 25 g/100 kg concentration, and then the *Saccharomyces* yeast at 30 g/100 kg, 3 days later. In the case of Melody, the inoculation was performed in one step at 30 g/100 kg as it is a mixture of yeasts. The tanks were sealed with air-tight covers, enabling the release of carbon dioxide. The fermentations were performed in triplicates at 16 ± 1 °C until no further changes were observed in the apparent extract.

### 2.3. Distillation Process

The fermented mash was distilled in a steam-heated still equipped with a rectifying column and dephlegmator (Hagyó Spirit Company, Miskolc, Hungary). The distillation unit was computer-controlled and process parameters including condenser temperature, reflux ratio, and heating program were set through software. The rectifying column was equipped with three bubble cap trays: the lower tray held 70%, the middle tray held 45%, while the upper tray was bearing only 15% condensate, which was flowing back as reflux from the dephlegmator (condenser). After completion of fermentation, all batches were distilled with the same distillation settings. Heart fractions were stored for two weeks before analysis.

### 2.4. Analytical Methods

Fermentation was monitored continuously by measuring the dry matter content (PAL-1 Refractometer, Atago, Tokyo, Japan), pH (FE20-Kit FiveEasy™ Benchtop pH Meter, Mettler Toledo, Greifensee, Switzerland), and the concentration of reducing sugars [11]. Titratable acidity was measured by potentiometric titration with 0.2 N NaOH, whereas the volatile acidity was quantified by steam distillation/titration with 0.1 N NaOH. After distillation, the ethanol content of the distillate was determined using the DMA 35N Portable Density Meter (Anton Paar, Graz, Austria).

### 2.5. Sugars and Organic Acids Analysis (HPLC)

The amounts of sugars and organic acids in the mash were determined by HPLC [12]. Briefly, the aliquot of the sample was centrifuged (14,000× *g* for 10 min). The supernatants were then filtered through a 0.45 µm membrane (Waters, Milford, MA, USA) and analyzed in triplicate by the Thermo Scientific Surveyor Plus HPLC System (Thermo Fisher Scientific, Waltham, MA, USA) consisting of an autosampler, Refractor Index (RI) and Photodiode Array (PDA) detectors, as well as a thermostatically controlled column compartment set at 45 °C. The ion exclusion column Aminex HPX-87H (BioRad, Hercules, CA, USA) was used with 5 mM H_2_SO_4_ as the eluent, with a flow rate of 0.5 mL/min. The data acquisition and integration were performed using the ChromQuest 5.0 software package (Thermo Fisher Scientific, Waltham, MA, USA). Standards of sugars (glucose, fructose, sucrose) and organic acids (lactic, acetic, succinic) were used to identify and quantify the components in the samples.

### 2.6. Volatile Compounds Analysis (GC-FID)

The selected volatile compounds formed during alcoholic fermentation were analyzed by GC-FID (PR 2100 Series Chromatography System, Perichrom, Paris, France). The separations were performed using a CP-WAX 57 CB (Agilent, Santa Clara, CA, USA) capillary column (50 m × 0.32 mm ID with 0.2 µm film thickness). The injector and detector temperatures were 220 and 240 °C, respectively. The following temperature program was established: 40 °C for 3 min at an increment of 6 °C/min to 75 °C, then 9 °C/min to 210 °C. The carrier gas was hydrogen at a 0.7 mL/min flow rate. External standards were used to identify and quantify the components in the sample. The concentrations of volatile compounds are provided in mg/L alcohol 100% *v/v*. All tests were performed in triplicates.

### 2.7. Sensory Analysis

Organoleptic properties of distilled spirits were evaluated using the 20-point scale test [13]. Four weeks before the evaluation, samples were diluted with distilled water to 43% (*v/v*) ethanol. The sensory evaluation was performed by a trained panel of 7 female and 8 male participants. The tasting procedure incorporated four criteria and a scale from 5 to 1. The criteria were as follows: Cleanliness (technological purity)—presence/absence of head and tail fractions and other technological defects (e.g., moldy mash, pickling), fruit character—the typical aroma of the distillate in terms of intensity and quality in the nose and on the palate, mouth-feel—examination of the flavors that can be felt in the mouth, their permanence, pleasantness, and elegance, and harmony—evaluation of overall impressions of the product and testing of the harmony of taste and smell.

### 2.8. Statistical Analysis

The mean values, the standard deviations of triplicate trials, as well as analysis of variances (ANOVA) followed by Tukey’s HSD test were performed using the SPSS software package (Version 13.0, SPSS Inc., Chicago, IL, USA).

## 3. Results and Discussion

### 3.1. Investigation of Fermentability of Different Starter Cultures

The analytical profiles of the fresh and fermented mashes obtained from pure and mixed fermentations are reported in Table 1. The fresh apple mash was characterized by a high total sugar content (148.3 g/L), which included reducing sugars with a concentration of 133 g/L. After completion of fermentation, in the mashes that were inoculated with mixed cultures Biodiva + Uva228 and Concerto + Uva228, lower concentrations of residual sugars were detected (10.6 and 11.3 g/L, respectively). This behavior highlights the high fermentation capacity of yeasts in mixed fermentation. The total acidity of the fresh mash was 5.3 g/L. However, following fermentation, this parameter increased by 1.6–2.3 g/L, owing to the synthesis of certain organic acids as normal products of yeast metabolism. In contrast, in the study of Satora et al. [14], a decreasing tendency of total acidity was shown in the plum mash after fermentation, which was probably a result of microbial activity. The co-inoculation Biodiva + Uva228 showed the lowest concentration of volatile acidity (0.33 g/L) compared with other samples. All mashes were characterized by a comparable consumption rate of sugars, 86.7–92.9%, whereas the highest ethanol production was observed in the fermentation with Uva228 (6 vol%).

### 3.2. Analyzed Sugars and Organic Acids Profile during the Fermentation Process

The amounts of sugars in the mash depend on the variety of fruit, climatic conditions, and time of harvest [14]. The apple mash was characterized by high initial concentrations of fructose (8.91 g/100 mL), glucose (4.06 g/100 mL), and sucrose (1.86 g/100 mL). All yeast strains showed similar patterns of sugar utilization (Figure 1). A sharper decrease in carbohydrate content was recorded in the first week of fermentation, indicating a more vigorous utilization rate of sugars. The fastest rate of fermentable sugars’ utilization was detected in the co-inoculation Biodiva + Uva228. No further decline in fermentable sugars’ content was observed after the 15th day, indicating the end of fermentation for all inoculum types. Similar decreasing trends in the concentration of sugars during the fermentation process were reported in the literature [9,14].

Figure 2 shows the evolving profiles of the main organic acids during fermentation. As illustrated in Figure 2A, lactic acid was formed throughout the fermentation process, with the final concentration being the highest in the co-inoculation Concerto + Uva288 and Melody. Lactic acid is synthesized by the reduction of pyruvic acid during glycolysis or the transformation of malic acid. Succinic acid is another common metabolite formed from pyruvic acid via malic acid, fumaric acid, and the decomposition of some amino acids. The importance of succinic acid is not solely due to its presence in the fruit mash, it also readily reacts with other molecules to form esters [15]. Its changing profile is shown in Figure 2B. The initial concentration of succinic acid in the apple mash was 0.27 g/L. After fermentation, its content increased sharply, with a minimum value of 1.9 g/L (Melody) and a maximum of 2.78 g/L (Uva228). Fluctuations in the concentration of acetic acid in mash were observed throughout the fermentation process, and the yeast strain used had a major influence on observed differences. However, final concentrations were similar among all tested samples, with an exception in the case of the co-inoculum Biodiva + Uva228, where the lowest concentration of acetic acid was detected (Figure 2C). The results were in agreement with previous studies [16].

### 3.3. Analyzed Volatile Compounds in the Apple Distillates

Volatile compounds of fruit distillates may originate from raw materials and may be formed as byproducts during alcoholic fermentation, distillation, and maturation [17]. Since our goal was to compare yeasts, we primarily focused on the aroma compounds produced during fermentation (Table 2). Esters, higher alcohols, acids, and acetaldehyde constitute the main group of compounds that make up the “fermentation bouquet”. Acetaldehyde is an important carbonyl compound found in alcoholic beverages, and in small concentrations, it has a fresh, “fruity” odor [18]. It is a metabolic product of the fermentation process, as well as chemical and enzymatic oxidation of ethanol [14]. The highest acetaldehyde concentration was noted in the sample fermented with Biodiva + Uva228 (199.32 mg/L), while in the other samples, the detected values were 125–152.34 mg/L. Winterová et al. [19] reported that the acetaldehyde content in apple brandies was in the range of 30–260 mg/L.

Higher alcohols are considered as the key aroma compounds in distillates. These alcohols (also known as fusel oils) are secondary yeast metabolites, formed during the alcoholic fermentation process from sugars and amino acids via the Ehrlich pathway. Excessive concentrations of higher alcohols can result in a harsh, pungent smell and taste, whereas optimal levels impart fruity characters [20]. Among analyzed higher alcohols, isoamyl alcohol predominated. Precursors for the formation of these alcohols could be leucine, acetic acid, and acetaldehyde [14,21]. The highest concentrations were found in the spirits produced with Uva228 (329.77 mg/L) and with Biodiva + Uva228 (297.4 mg/L), and the lowest with Concerto + Uva228 (209.87 mg/L). The other samples were characterized by a fairly uniform level of this compound (227.07–243.61 mg/L). Rusu Coldea et al. [22] measured isoamyl alcohol values between 75.28 and 196.59 mg/100 mL in different apple brandies.

In addition to isoamyl alcohol, high amounts of 1-propanol (157.32–206.1 mg/L) and 2-methyl-1-butanol (88.45–149.47 mg/L) were detected in the samples. 1-Propanol has a pleasant, sweetish odor, but excessive concentrations will introduce solvent notes that mask all the positive notes in distillates [20]. The highest concentration of this compound was observed in the sample made with Biodiva + Uva228 (206.1 mg/L), and the lowest in the sample Uva228 (157.3 mg/L). Nearly similar amounts of 1-propanol were measured in cherry (132–300 mg/L) and plum (166–303 mg/L) distillates [14,20]. In the case of Biodiva and Melody starter cultures, the 2-methyl-1-butanol content of less than 100 mg/L was detected. This compound showed the highest value (149.47 mg/L) in the sample Concerto + Uva228. The quantities of the other higher alcohols were lower in the investigated samples. The shares of 1-hexanol, 2-phenylethanol, 1-butanol, 2-butanol, *trans*-3-hexen-1-ol, *cis*-2-hexen-1-ol, and benzyl alcohol accounted for less than 10% of the total amount of the higher alcohols. Among these, the largest quantities of 1-hexanol (47.62 mg/L) and 2-phenylethanol (29.84 mg/L) were detected in the distillate fermented with Concerto + Uva228. 1-Hexanol is not a fermentation product, but most often originates from linolenic acid found in the green parts of plants and unripe fruits [14]. 2-Phenylethanol has a positive influence on the aroma of the distillate and is derived from L-phenylalanine through the metabolic reaction of yeast during carbonic anaerobiosis [20]. This compound was not detected in Biodiva and Concerto samples. All samples were characterized by low amounts of 1-butanol (3.38–4.09 mg/L). No significant differences in 1-butanol production were observed between strains. The compound 2-butanol was not detected in Biodiva and Concerto samples. Spaho et al. [23] mentioned that the presence of 2-butanol is related to bacterial action and an amount of 7–8 mg/100 mL ethanol is a guarantee of fermentation. The two aliphatic alcohols, 3-hexen-1-ol and *cis*-2-hexen-1-ol, originate from the process of crushing and maceration of fruits. The highest concentration of 3-hexen-1-ol was measured in the samples fermented with Concerto + Uva228 and Melody.

Esters are formed during alcoholic fermentation via yeast metabolism and qualitatively present the major class of flavor compounds in distillates. Esters contribute to the pleasant fruity aroma of fruit distillates [23]. The most abundant ester was ethyl acetate. In low concentrations (up to 200 mg/L), it has a floral and fruity aroma. At higher concentrations, it has a negative impact on the sensory quality of spirits [14]. Analyzed samples were characterized by diversified content of ethyl acetate, ranging from 131.60 mg/L in Concerto to 195.5 mg/L in Melody. Ethyl hexanoate supplies the aroma of fruit (banana, green apple, etc.), and its presence, along with other ethyl esters, is beneficial for the spirit [20]. The highest content of this compound was observed in sample Uva228 (6.026 mg/L) and the lowest in Biodiva (3.891 mg/L). In addition to ethyl hexanoate, significant amounts of ethyl octanoate and ethyl benzoate were measured in samples. These compounds were present in higher amounts in the samples fermented with Biodiva and Melody. Phenylethyl acetate, isoamyl acetate, and propyl acetate were present in very low concentrations in the analyzed spirits. Furthermore, propyl acetate was not detected in three samples (Uva228, Biodiva, Biodiva + Uva228). A similar result was observed for ethyl formate. Methanol production is associated with the enzymatic degradation of the methoxy groups of pectin, as well as the acidic degradation of pectin [20]. Methanol does not directly influence the flavor of the distillate; however, it is subjected to restrictive controls due to its high toxicity [24]. The methanol content in the analyzed samples ranged between 1706.07 and 1986.88 mg/L (maximum legal limit is 10,000 mg/L of 100% vol. ethanol) [25]. The linalool profile was similar in all distillates.

### 3.4. Sensory Analysis

The results of the sensory evaluations are provided in Table 3. The total scores ranged between 15.2 (Melody) and 18.9 (Concerto + Uva228). All samples received a maximal score for technological purity, indicating that the hearts were properly cut from head and tail fractions during the distillation process. The fruitiness and high flavor intensity perceived by the panelists were highly appraised, especially in the distillates produced from the mixed inoculums (Concerto + Uva228 and Biodiva + Uva228). The results obtained in the sensory analysis could be correlated with those obtained from the chemical characterization (Table 2 and Table 3). The use of a mixed inoculum of *S. cerevisiae* and non-*Saccharomyces* yeasts enabled the production of more esters (which provide the sweet taste and smell of fruit and flowers) and higher alcohols (which provide the taste and smell of coconut and honey and the smell of roses) [9]. The best sensory outcomes of fruit distillates are a consequence of good balances of the quantities of aromatic compounds. In our study, the best-rated distillate was the one produced by the mixed culture Concerto + Uva228. This sample was characterized by a pleasant, delicate apple aroma (fresh fruit with a citrus-like, skin spicy aroma) and a well-harmonized, refreshing, pleasantly burning taste.

## 4. Conclusions

Analyzed byproducts’ concentration, an efficient sugar utilization, and a reduction in volatile acidity support the fact that non-*Saccharomyces* yeasts bring suitable enological characteristics to Pálinka. An increase in ester and higher alcohol content shows that non-*Saccharomyces* yeasts distinctively modulate the concentrations of specific fermentative volatiles, thus highlighting the fruity and floral traits in the distillate. The present study provides a promising strategy to improve the overall quality of Pálinka, and this statement is also supported by the results of sensory analysis, where the panelists mostly favored the distillates produced by non-*Saccharomyces* yeasts (*T. delbrueckii* and *L. thermotolerans*) in single or mixed/sequential fermentations.

## Figures and Tables

**Figure 1 foods-10-01336-f001:**
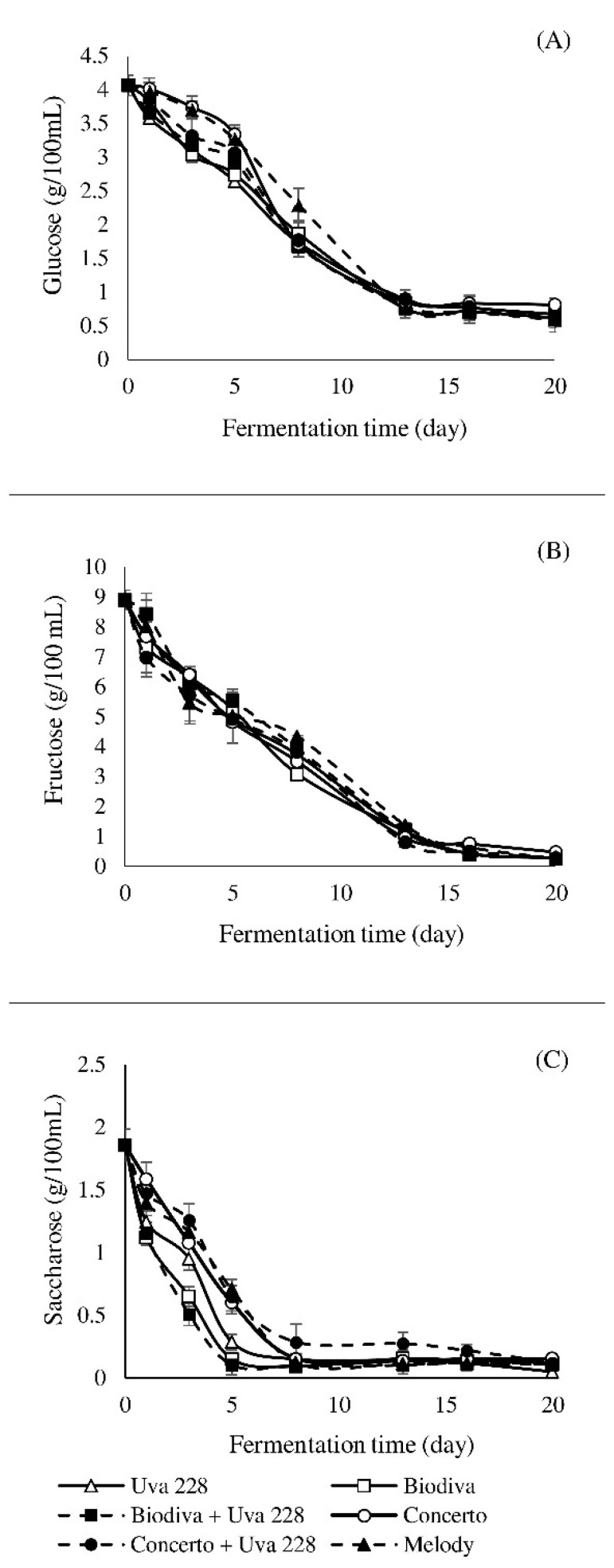
The concentrations of glucose (**A**), fructose (**B**), and saccharose (**C**) in apple mash during the fermentation process.

**Figure 2 foods-10-01336-f002:**
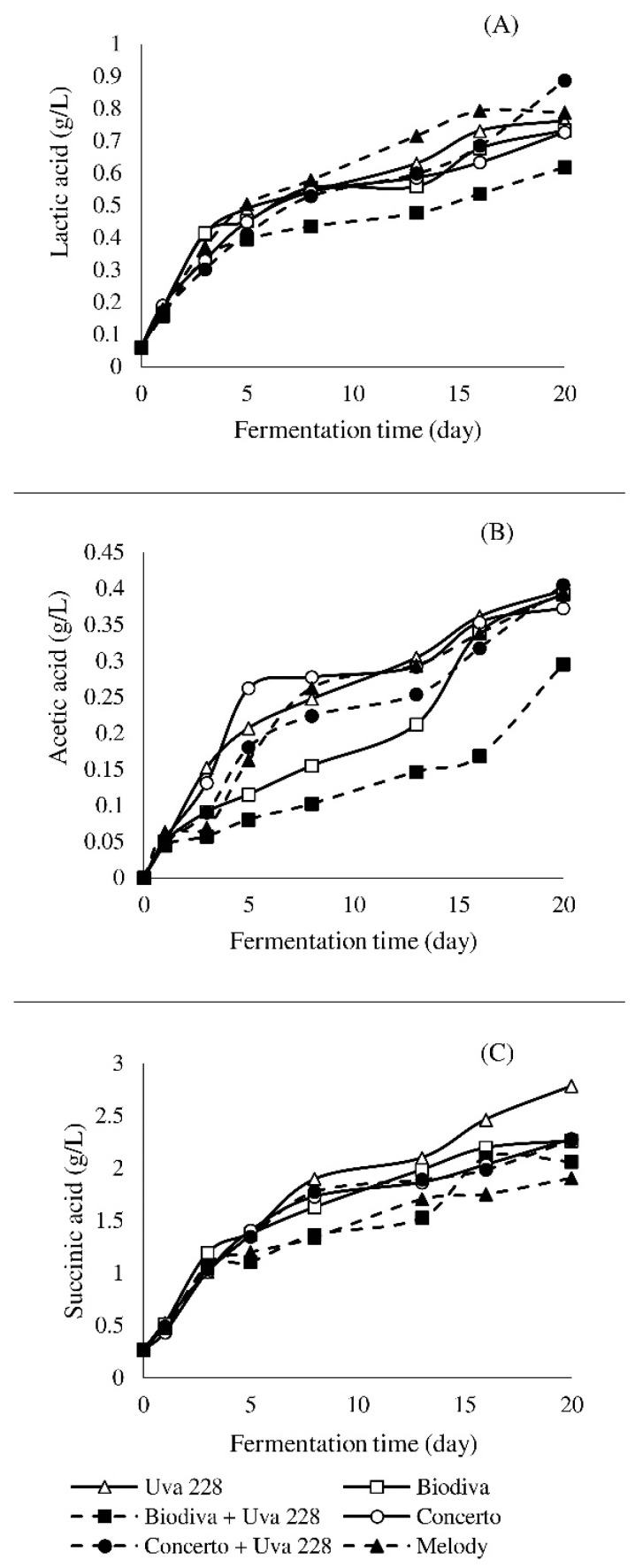
The concentrations of lactic acid (**A**), acetic acid (**B**), and succinic acid (**C**) in apple mash during the fermentation process.

**Table 1 foods-10-01336-t001:** The main enological parameters of fresh and fermented apple mashes.

	Refraction (*w/w*%)	Total Sugars (g/L)	Reducing Sugars (g/L)	Titratable Acidity (g/L)	pH	Volatile Acidity (g/L)	Ethanol (vol%)	Sugars’ Consumption (%)
**Fresh apple mash**	15.8 (±0.5)	148.3 (±6.2)	133 (±5.1)	5.3 (±0.34)	3.58 (±0.12)	n.a.	n.a.	n.a.
**Fermented mash**								
Uva228	5.2 a (±0.17)	11.8 a (±2.1)	11.2 a (±2.3)	7.6 b (±0.34)	3.19 a (±0.12)	0.50 a (±0.08)	6.00 d (±0.08)	92
Biodiva	5.3 ab (±0.18)	13.3 a (±1.1)	12.1 a (±1.5)	7.3 ab (±0.19)	3.17 a (±0.08)	0.42 a (±0.11)	5.20 b (±0.10)	91
Biodiva + Uva228	5.1 a (±0.22)	10.6 a (±1.3)	9.5 a (±2.2)	7.2 ab (±0.23)	3.10 a (±0.09)	0.33 a (±0.09)	5.60 c (±0.15)	92.9
Concerto	5.7 b (±0.11)	19.7 b (±1.7)	18.2 b (±2.4)	7.1 ab (±0.17)	3.16 a (±0.15)	0.45 a (±0.05)	4.80 a (±0.10)	86.7
Concerto + Uva228	5.25 ab (±0.2)	11.3 a (±2.1)	10.1 a (±1.2)	6.9 a (±0.13)	3.14 a (±0.10)	0.50 a (±0.10)	5.70 cd (±0.12)	92.4
Melody	5.2 a (±0.23)	13.5 a (±1.9)	12.5 a (±2.5)	6.9 a (±0.25)	3.15 a (±0.13)	0.36 a (±0.06)	5.60 c (±0.20)	90.9

Data are expressed as mean ± standard deviation; n.a.: not analyzed. Values with different letters (a–d) in the same column are significantly different according to Tukey’s HSD test (*p* < 0.05).

**Table 2 foods-10-01336-t002:** Volatile aroma compounds identified in the apple distillates.

Compound (mg/L Alcohol 100% *v/v*)	Uva228	Biodiva	Biodiva + Uva228	Concerto	Concerto + Uva228	Melody
Methanol	1706.07 a (±125.56)	1710.03 a (±115.62)	1720.68 a (±134.25)	1986.88 a (±142.85)	1933.19 a (±147.21)	1944.73 a (±131.51)
Acetaldehyde	140.49 a (±17.23)	133.82 a (±12.56)	199.32 b (±15.67)	152.34 a (±14.38)	125.00 a (±7.45)	125.49 a (±9.89)
Isoamyl alcohol	329.77 c (±32.27)	241.17 ab (±27.49)	297.40 bc (±21.91)	227.07 ab (±22.63)	209.87 a (±18.02)	243.61 ab (±31.71)
1-Propanol	157.32 a (±11.53)	172.77 a (±17.78)	206.10 a (±19.66)	176.12 a (±10.34)	167.73 a (±22.24)	163.78 a (±13.23)
1-Butanol	3.639 a (±0.56)	3.380 a (±0.34)	3.747 a (±0.37)	4.097 a (±0.16)	3.828 a (±0.35)	3.971 a (±0.23)
2-Butanol	0.293 a (±0.031)	n.d.	0.604 c (±0.049)	n.d.	0.428 b (±0.022)	0.754 d (±0.052)
1-Hexanol	33.12 ab (±2.46)	27.35 a (±3.04)	34.23 b (±2.14)	43.56 c (±3.56)	47.62 c (±1.78)	46.86 c (±4.16)
2-Phenylethanol	22.44 b (±1.64)	n.d.	13.24 a (±0.96)	n.d.	29.84 c (±2.06)	27.02 c (±1.87)
2-Methyl-1-butanol	110.56 ab (±9.14)	88.45 a (±7.52)	116.89 b (±11.64)	115.08 b (±12.44)	149.47 c (±15.02)	95.50 ab (±8.34)
trans-3-Hexen-1-ol	0.043 ab (±0.005)	0.018 a (±0.003)	0.061 bc (±0.010)	0.080 c (±0.004)	0.154 d (±0.012)	0.136 d (±0.011)
cis-2-Hexen-1-ol	0.017 a (±0.002)	0.017 a (±0.001)	0.019 a (±0.003)	0.020 a (±0.003)	0.018 a (±0.002)	0.023 a (±0.003)
Benzyl alcohol	0.420 cd (±0.022)	0.470 d (±0.055)	0.370 cd (±0.050)	0.120 a (±0.011)	0.230 b (±0.033)	0.335 c (±0.031)
Ethyl acetate	178.50 bc (±10.35)	147.30 ab (±25.46)	165.20 abc (±12.26)	131.60 a (±10.67)	167.40 abc (±14.24)	195.50 c (±15.03)
Propyl acetate	n.d.	n.d.	n.d.	0.016 a (±0.003)	0.012 a (±0.002)	0.026 b (±0.002)
Ethyl hexanoate	6.026 c (±0.35)	3.891 a (±0.27)	4.638 ab (±0.47)	4.286 ab (±0.56)	4.024 ab (±0.36)	4.928 b (±0.28)
Ethyl butyrate	0.022 a (±0.002)	0.054 b (±0.006)	0.043 b (±0.003)	0.049 b (±0.004)	0.045 b (±0.006)	0.055 b (±0.010)
Isoamyl acetate	0.033 ab (±0.006)	0.042 bc (±0.005)	0.056 c (±0.011)	0.018 a (±0.002)	0.027 ab (±0.003)	0.057 c (±0.010)
Phenylethyl acetate	0.044 a (±0.005)	0.025 b (±0.010)	0.055 ab (±0.006)	0.039 a (±0.006)	0.053 ab (±0.012)	0.045 ab (±0.006)
Diethyl succinate	0.245 a (±0.045)	0.412 c (±0.035)	0.295 ab (±0.025)	0.371 bc (±0.023)	0.387 c (±0.033)	0.378 c (±0.032)
Ethyl octanoate	3.045 bc (±0.34)	3.425 c (±0.55)	2.962 bc (±0.24)	2.130 a (±0.17)	2.450 ab (±0.12)	2.794 abc (±0.31)
Ethyl benzoate	3.805 b (±0.64)	4.467 c (±0.21)	4.079 bc (±0.34)	1.560 a (±0.14)	2.274 a (±0.27)	4.773 cd (±0.34)
Ethyl formate	n.d.	n.d.	n.d.	0.563 a (±0.051)	0.693 b (±0.066)	0.633 ab (±0.025)
Linalool	0.117 ab (±0.015)	0.169 bc (±0.021)	0.129 ab (±0.017)	0.182 c (±0.031)	0.150 abc (±0.022)	0.105 a (±0.011)

Data are expressed as mean ± standard deviation; n.d.: not detected. Values with different letters (a–d) in the same row are significantly different according to Tukey’s HSD test (*p* < 0.05).

**Table 3 foods-10-01336-t003:** Sensory analysis of apple spirits obtained from different starter cultures.

	Technology Purity (Max 5 Points)	Fruit Character (Max 5 Points)	Mouthfeel (Max 5 Points)	Harmony (Max 5 Points)	Total (Max 20 Points)
Uva228	5 (±0)	3.93 (±0.46)	3.73 (±0.59)	3.46 (±0.63)	16.1 (±1.24)
Biodiva	5 (±0)	4.33 (±0.61)	4.27 (±0.59)	4.4 (±0.63)	18 (±1.55)
Biodiva + Uva228	5 (±0)	4.26 (±0.59)	3.93 (±0.46)	3.73 (±0.46)	16.9 (±0.79)
Concerto	5 (±0)	3.8 (±0.56)	3.8 (±0.67)	3.4 (±0.73)	16 (±1.36)
Concerto + Uva228	5 (±0)	4.86 (±0.35)	4.46 (±0.52)	4.6 (±0.5)	18.9 (±1.03)
Melody	5 (±0)	3.53 (±0.64)	3.46 (±0.74)	3.2 (±0.77)	15.2 (±1.69)

Data are expressed as mean ± standard deviation.

## Data Availability

The data presented in this study are available upon request from the corresponding author. The data are not publicly available due to privacy.

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
