# Peer review of "Influence of Non-Saccharomyces Strains on Chemical Characteristics and Sensory Quality of Fruit Spirit"

_foods, 2021, doi:10.3390/foods10061336_

Round 1
Reviewer 1 Report
Dear Authors,
my compliments for your work.
Some minor revisions are necessary:
Section 2.1 Please list the external standards purchased from Sigma-Aldrich and used to identify and quantified in the sample.
Section 2.6 Row 105 the “ selected volatile compounds “ please specify why you have selected these compounds in particular.
Best regards
Author Response
Dear Authors, my compliments for your work. Some minor revisions are necessary.
Dear Reviewer, thank you for your kind words. We appreciate your careful review and constructive suggestions. We tried to carefully revise our manuscript based on your comments.
Point 1: Section 2.1 Please list the external standards purchased from Sigma-Aldrich and used to identify and quantified in the sample.
Response 1: In Section 2.1. Raw Materials, Yeast Strains, and Chemicals; we listed all the chemicals obtained from Sigma-Aldrich, used as external standards during instrumental analysis (HPLC and GC) to identify and quantify the components in the samples (line 60-66).
Point 2: Section 2.6 Row 105 the “selected volatile compounds “ please specify why you have selected these compounds in particular.
Response 2: In Section 2.6. Volatile Compounds Analysis (GC-FID), row 105, the sentence has been modified to clarify and highlight why the examined volatile compounds were chosen (previously, line 105; in the revised manuscript, line 117). Furthermore, in Section 3.3. Analyzed Volatile Compounds in the Apple Distillates, the written statement supporting the reason behind the investigation of certain volatile compounds was enriched to make it even more comprehensible and to better frame the context of our study (previously, line 181-184; in the revised manuscript, line 198-202).
It is our belief that the manuscript is substantially improved after making the suggested edits. We would be happy to make any further changes that may be required.
Reviewer 2 Report
It is a fairly straightforward fermentation manuscript where wine non-Saccharomyces yeast were used to co-ferment the apple mash.
Some comments:
Big issue: There is a clear lack of statistical analysis in Table 1 and 2. The different fermentations are not adequately compared with each other. This needs to be corrected before this manuscript can be accepted.
Minor issues: Kluyveromyces thermotolerans has been reclassified and is now Lachancea thermotolerans
line 52: "domestica" should be a small letter?
line 54-55: combine the two sentences
line 90-91: the "n" should be capitalized
line 120,123: remove the etc.
Figure 1: Perhaps better to make the three figures equal in size beneath each other.
line 168: the reference (15) did not change
Author Response
It is a fairly straightforward fermentation manuscript where wine non-Saccharomyces yeast were used to co-ferment the apple mash.
Dear reviewer, we would like to thank you for your careful and thorough reading of our manuscript and for the thoughtful comments and constructive suggestions, which help to improve the quality of this manuscript. We have incorporated the suggested changes into the manuscript to the best of our ability.
Point 1: There is a clear lack of statistical analysis in Table 1 and 2. The different fermentations are not adequately compared with each other. This needs to be corrected before this manuscript can be accepted.
Response 1: We agree with your assessment, so we have performed additional statistical analysis to better represent the results obtained from the conducted experiment. Statistically significant differences between each results (p < 0.05) were evaluated using One-way ANOVA with Tukey HSD Test. Each group is indicated by Roman letters in Table 1 and 2.
Point 2: Kluyveromyces thermotolerans has been reclassified and is now Lachancea thermotolerans
Response 2: The specie name “Kluyveromyces thermotolerans” has been modified to “Lachancea thermotolerans” throughout the manuscript.
Point 3: line 52: "domestica" should be a small letter?
Response 3: The correction has been made. The word “Domestica” has been changed to “domestica” (in the revised manuscript, line 52).
Point 4: line 54-55: combine the two sentences
Response 4: The sentence has been rewritten as suggested (in the revised manuscript, line 54-56).
Point 5: line 90-91: the "n" should be capitalized
Response 5: In section 2.4. Analytical Methods, the “n” has been capitalized to “N” (in the revised manuscript, line 99-100).
Point 6: line 120,123: remove the etc.
Response 6: The correction has been made as suggested, the “etc.” has been removed (in the revised manuscript, line 134,136).
Point 7: Figure 1: Perhaps better to make the three figures equal in size beneath each other.
Response 7: Formatting of Figure 1 has been done as suggested. Since Figure 2 had the same structure in the original manuscript, we modified both of them to meet the suggested format.
Point 8: line 168: the reference (15) did not change
Response 8: In section 3.2. Analyzed Sugars and Organic Acids Profile during the Fermentation Process, in order to avoid repeating the same reference [15], the text has been revised and modified (previously, line 162-168; in the revised manuscript, line 178-184).
It is our belief that the manuscript is substantially improved after making the suggested edits. We would be happy to make any further changes that may be required.
Reviewer 3 Report
Dear Authors. I read Your manuscript very carefully and I agree that the topic is very interesting and the use of non S.cerevisiae strains in preparation of fruit distillates is still not well estimated. The methods used by You are standard methods used in such kinds of investigations, obtained results are clearly presented, and were compared with others' results. I was able to spot only one editorial imperfections which , in my opinion, should be removed - free spaces in table 1 ( in the line of unfermented mash results)- please see my comment in the enclosed file.

Author Response
Point 1: Dear Authors. I read Your manuscript very carefully and I agree that the topic is very interesting and the use of non S.cerevisiae strains in preparation of fruit distillates is still not well estimated. The methods used by You are standard methods used in such kinds of investigations, obtained results are clearly presented, and were compared with others' results. I was able to spot only one editorial imperfections which, in my opinion, should be removed - free spaces in table 1 (in the line of unfermented mash results) - please see my comment in the enclosed file.
Dear reviewer, thank you for the appreciation of our work. We would like to thank you for the time and effort that you have dedicated to providing your valuable feedback on our manuscript.
Response 1: Following your suggestion, Table 1 has been revised. The abbreviation “n.a.” was used to replace the empty spots in the second row of the table, where no analysis was relevant for the Fresh apple mash, while the explanation “n.a.: not analyzed” is supplied in the table footer.
We would be happy to make any further changes that may be required.